# Comprehensive Review on g-C_3_N_4_-Based Photocatalysts for the Photocatalytic Hydrogen Production under Visible Light

**DOI:** 10.3390/ijms24010346

**Published:** 2022-12-25

**Authors:** Angelina V. Zhurenok, Danila B. Vasilchenko, Ekaterina A. Kozlova

**Affiliations:** 1Federal Research Center Boreskov Institute of Catalysis, Siberian Branch of the Russian Academy of Science, Pr. Ak. Lavrentieva, 5, Novosibirsk 630090, Russia; 2Nikolaev Institute of Inorganic Chemistry, Siberian Branch of the Russian Academy of Science, Novosibirsk 630090, Russia

**Keywords:** g-C_3_N_4_, hydrogen, photocatalysis, platinum

## Abstract

Currently, the synthesis of active photocatalysts for the evolution of hydrogen, including photocatalysts based on graphite-like carbon nitride, is an acute issue. In this review, a comprehensive analysis of the state-of-the-art studies of graphic carbon nitride as a photocatalyst for hydrogen production under visible light is presented. In this review, various approaches to the synthesis of photocatalysts based on g-C_3_N_4_ reported in the literature were considered, including various methods for modifying and improving the structural and photocatalytic properties of this material. A thorough analysis of the literature has shown that the most commonly used methods for improving g-C_3_N_4_ properties are alterations of textural characteristics by introducing templates, pore formers or pre-treatment method, doping with heteroatoms, modification with metals, and the creation of composite photocatalysts. Next, the authors considered their own detailed study on the synthesis of graphitic carbon nitride with different pre-treatments and respective photocatalysts that demonstrate high efficiency and stability in photocatalytic production of hydrogen. Particular attention was paid to describing the effect of the state of the platinum cocatalyst on the activity of the resulting photocatalyst. The decisive factors leading to the creation of active materials were discussed.

## 1. Introduction

Fossil fuels are a limited and exhaustible resource for conventional energy production that induces reasonable concerns of energy and economic crises [1]. Besides, wide use of fossil fuels causes environmental and human health impacts [2]. In this context, triggering hydrogen as an alternative energy source is a promising trend, since hydrogen is an environmentally benign fuel [3,4,5]. Currently, numerous industrial processes to produce hydrogen are available, but they all are energy intensive, require high temperatures, and appear economically efficient only at large scales [6]. In this regard, the photocatalytic production of hydrogen attracts particular interest as the process proceeding at ambient conditions and simulating photosynthesis, i.e., the direct conversion of solar energy into the energy of chemical bonds [7,8,9,10,11,12,13,14,15,16].

The process of photocatalytic splitting of water using TiO_2_ catalysts under UV radiation was pioneered by Fujishima and Honda in 1972 [17]. Since that time, a great number of research has been performed to produce hydrogen by photocatalytic splitting of water with various semiconductor photocatalysts [18,19,20,21,22]. However, this process has an intrinsic problem of hydrogen-oxygen recombination [23]. To improve the process’ quantum efficiency, photocatalytic reduction of water is carried out with the use of electron donor agents, such as various organic and inorganic compounds capable of donating electrons and thereby reducing the recombination of electron-hole pairs on the photocatalyst surface [24,25,26].

Promising photocatalysts activated by UV and visible light include TiO_2_ [27,28,29,30], ZnO [31], Fe_2_O_3_ [32], CdS [33], Cd_1−_Zn_x_S [34], Bi_2_WO_6_ [35], BiVO_4_ [36], Ta_2_O_5_ [37], Ta_3_N_5_ [38], and TaON [39]. Nowadays, the research and development of high-performance semiconductor photocatalysts for solving the problems of energy shortages and environmental safety are particularly important. Recently, visible-light-activated photocatalysts have been developed, which allow efficient use of the solar spectrum containing a large fraction of visible light (about 43%) [17]. Traditional semiconductor photocatalysts, such as TiO_2_, have a large band gap and, therefore, are unable to absorb visible light, and can be activated only by UV light, which constitutes a small fraction of the solar radiation spectrum [40,41,42]. Extensive searches for reliable visible-light-activated semiconductor photocatalysts revealed the next generation photocatalyst based on a polymeric semiconductor—graphitic carbon nitride [43,44,45,46].

## 2. Graphitic Carbon Nitride as a Photocatalyst

Carbon nitride, albeit known for a long time, has been tested as a photocatalyst only since 2009 [47]. Every year, the interest of researchers in this material increases, as shown in Figure 1a. It can also be seen in Figure 1b that, to date, about 10–12% of all studies on the photocatalytic H_2_ production devoted to the synthesis and study of photocatalysts based on g-C_3_N_4_ and their applications.

It reveals various allotropic forms, such as α-C_3_N_4_, β-C_3_N_4_, graphitic, cubic, and pseudocubic [48]. The most stable modification is a polymeric graphitic structure in which s-triazine or tri-s-triazine (s-heptazine) units are bound to each other through tertiary amines [47]. Graphitic carbon nitride g-C_3_N_4_ has the band gap of 2.7 eV and the conduction band-edge potential −1.3 V vs. NHE, which enables an efficient hydrogen production process [47]. This material is thermally and chemically stable, safe for the environment, acid and alkali resistant; its surface can be modified without infringing its composition and structure [49]. It is easily synthesized by thermal polycondensation of inexpensive nitrogen-containing precursors, such as dicyandiamide, cyanamide, melamine, urea, and thiourea [46,50,51,52]. The downside of this process is that it yields materials with low specific surface areas and high electron-hole recombination rates that provokes the loss in the catalytic activity [53]. To stabilize the photocatalytic activity of g-C_3_N_4_, the following approaches are used:Alteration of textural characteristics by introducing templates, pore formers, or pretreatment method [54,55,56,57,58,59,60,61,62,63,64,65,66];Doping with heteroatoms [67,68,69,70,71,72,73];Modification with metals [74,75,76,77,78,79,80,81,82,83,84,85,86,87,88,89,90];Creation of composite photocatalysts [91,92,93,94,95,96,97,98,99,100].

The formation of graphitic carbon nitride proceeds by various routes at thermal condensation of the abovementioned precursors. At thermal pyrolysis, the cyclization of any of the above mentioned nitrogen-containing precursors yields melamine [49]. The dimerization of melamine at 350 °C yields melem and melon that transforms into polymeric g-C_3_N_4_ at temperatures above 500 °C (Figure 2).

Graphitic carbon nitride is known to have a defect-rich structure, which results from incomplete deamination during thermal polycondensation [49]. For this reason, synthesizing high-crystalline g-C_3_N_4_ is a difficult task, albeit exact crystallinity (correct arrangement of atoms) is a critically important parameter affecting the band structure and charge recombination rate (e^ȡ^/h^+^) of a photocatalytically active material. A large number of structural defects increases the charge recombination rate, and thus impedes photocatalyst activity. Texture controlling can improve the chemical, physical, and optical properties of g-C_3_N_4_ [101]. The relationship between the g-C_3_N_4_ morphology and photocatalytic hydrogen evolution performance is discussed below.

### 2.1. Textural Characteristics and Methods to Control Them

#### 2.1.1. The Use of Templates

The textural and morphological characteristics, such as porosity, specific surface area, and pore size distribution, of a carbon nitride based material can be controlled by using appropriate templates, which are subdivided into hard and soft ones. Hard templates are, for example, SiO_2_ and Al_2_O_3_ [56,57]. Graphitic carbon nitrides, synthesized using these templates, possess high specific surface area and porosity, and a large number of the surface active sites. Nevertheless, soft templates, such as starch and glucose, seem much more attractive, as they are safe for the environment, require no corrosive chemicals for elimination [58,59], and provide g-C_3_N_4_ with a specific surface area of >60 m^2^·g^−1^ [58]. After elimination of solvents and templates, the g-C_3_N_4_ nanoporous structure turns quite delicate (fragile) and easily destroyable, which presents one of the main problems of creating polymeric hollow nanospheres. A green soft-template approach allows synthesizing various g-C_3_N_4_ structures through relatively simple routes, keeping these structures durable even after template removal. Amphiphilic molecules, surfactants, ionic liquids, and gas bubbles are used as soft templates [60]. However, the template polymeric materials are inclined to early decomposition that results in some carbon residue in the material, which induces pores isolating, and thus affects negatively the g-C_3_N_4_ structure and photocatalytic activity.

#### 2.1.2. The Use of Pore-Forming Agents

Pore-forming agents can significantly improve the photocatalytic characteristics of graphitic carbon nitride due to their affecting of its textural characteristics. Ammonium chloride [61] and urea [62] are among the most commonly used pore-forming agents reported in the literature. This approach ensures the synthesis of complex porous structures, such as ultrathin nanosheets [61]. Favorably compared to hard templates, the pore-forming agents require no eliminating from the target material by using harmful compounds or procedures. For example, ammonium chloride (NH_4_Cl) additive in the g-C_3_N_4_ precursor decomposes at temperatures of 280–370 °C into NH_3_ and HCl thus facilitating the formation of a loose ultrathin nanosheet structure (thickness 1–2.4 nm or 3–7 atomic layers) with a defective matrix [61,63]. The bottleneck of this process is the cost-efficiency and ecological issues; obviously, the cheaper and environmentally benign pore-forming agents should be developed.

#### 2.1.3. Pretreatment Method

There are many studies in which precursors are modified prior to calcination in order to obtain g-C_3_N_4_ with improved properties [64,65,66]. One such modification method is the pretreatment of the precursor with acids (H_2_SO_4_, HNO_3_), which makes it possible to obtain a much higher specific surface area of resulting g-C_3_N_4_ [65]. The strong oxidizing power of the acids separates the g-C_3_N_4_ layers into thinner nanosheets, weakening the van der Waals force between them [65]. The addition of strong alkalis (NaOH) also facilitates the creation of ultrathin nanosheets. However, despite the ability to improve textural characteristics, it is difficult to control the yields and reproducibility of the resulting material in these pretreatment methods [65].

### 2.2. Doping Heteroatoms

The polymeric structure of graphitic carbon nitride is suitable for doping by other atoms or molecules by means of appropriate chemical procedures. The introduction of various atoms affects the electronic structure of the polymer material. Doping with nonmetals or anions leads to the g-C_3_N_4_ band gap narrowing. It is assumed that the band gap narrowing results from the formation of localized states and elevation of the valence band maximum due to the presence of doping atoms [67]. Besides doping with mono-nonmetal heteroatoms, co-doping with several heteroatoms is applied to provide more efficient band gap width variation [68,69,70].

Particular attention is focused on the doping of graphitic carbon nitride with halogens: Br, I, Cl [71,72]. For example, doping with bromine increases optical absorption, conductivity, charge carrier transfer rate, and photocatalytic activity without disturbing the g-C_3_N_4_ structural stability. It is assumed that halogen doping shifts the optical absorption spectrum towards longer wavelengths in the UV-visible spectrum [72]. The hindering of the charge carrier recombination at the introduction of Br atoms is attributed to the delocalization of the Br valence electrons to π-conjugated g-C_3_N_4_ structures. Graphitic carbon nitride doped with halogen atoms (F, Cl, Br, I) exhibited enhanced optical absorption and improved photocatalytic characteristics [72]. It was found that electronegativity of the halogen atoms affects the photocatalytic properties of g-C_3_N_4_. The higher the halogen atomic number, the lower is its electronegativity, and more electrons can transfer from the halogen atom to g-C_3_N_4_, thus leading to the gradually uplifting Fermi level and decreasing work function. Doping halogen atoms facilitate the electron escape from the g-C_3_N_4_ surface to participate in photocatalytic reactions.

Doping is an effective approach to improve the photocatalyst light absorption capacity, oxidizability, and separation efficiency of photoinduced charge carriers [73].

### 2.3. Doping Metals

One of the most frequently used and quite simple methods to modify the surface of graphitic carbon nitride is the use of metal dopants. The nanoparticles of noble and non-noble metals improve photocatalyst ability to absorb visible light and resist recombination of photoinduced charges. Also, metal nanoparticles can act as catalysts for the formation of molecular hydrogen. Noble metals attract particular attention, as they have the most suitable electronic and optical characteristics [74,75]. The Schottky barrier, also known as the space charge separation region, is formed at the noble metal-semiconductor heterojunction; it impedes electron migration from one material to another, and reduces the charge recombination. The creation of the Schottky barrier allows for the accumulation of additional negative charges (electrons) and positive charges (holes) in the noble metal and semiconductor, respectively. Noble metals also have the ability to absorb visible light owing to the plasmon resonance effect. There are two main approaches for depositing noble metal particles on the catalyst surface: impregnation with a precursor followed by reduction with NaBH_4_ [76] or another reductant [83], and photodeposition under the action of ultraviolet radiation [77,84]. There is also a possibility to attach pre-synthesized nanocrystals of desired sizes and shapes on graphitic carbon nitride nanosheets through electrostatic attraction [85]. The chlorocomplexes are common precursors for the deposition of noble metals onto a g-C_3_N_4_ surface [86,87,88]: for example, hexachloroplatinic acid (H_2_PtCl_6_) is usually used as the platinum precursor [78,84,89]. The photodeposition method does not use high-temperature treatments or hazardous materials. The downside of using noble metals as cocatalysts is that they are very expensive, especially in view of the high metal loading in the catalysts (1–5 wt.%) [79]. Currently, many studies are aimed at replace noble metals with non-noble ones, such as copper or nickel, which are much less expensive and are able to absorb visible light [80,81,90].

Although metal doping facilitates significant acceleration of the hydrogen production reaction, an excess of cocatalyst on the surface reduces the catalyst ability to absorb light in the visible region and acts as an electron trap [82].

### 2.4. Development of Composite Photocatalysts

In order to provide efficient light utilization and enhanced the redox ability of a material, it is common practice to shift the energy levels of the valence and conduction bands by creating a composite of two different materials. Composite materials are used to develop photocatalysts with enhanced light absorption in the visible region, efficient charge separation retarding charge recombination, and improved redox performance. There are various composite photocatalysts differing by heterojunction types: type I [91], type II [91,92], S-scheme [93,94], Z-scheme [95,96,97].

The creation of effective composites between two different compounds depends on their crystal structures, electron affinities, band structures, and the strength of the interfacial interaction. The type I heterojunctions assume that the valence band (VB) of semiconductor 1 lies higher than the VB of semiconductor 2, and the conduction band (CB) lies lower; in such a system, both electrons and holes are transferred to one semiconductor, where redox reactions occur [91]. Such a transfer of photogenerated charges from one semiconductor to another does not improve the efficiency of charge separation.

In type II heterojunctions, the level of VB and CB potentials of semiconductor 1 are higher than the level of VB and CB potentials of semiconductor 2. Here, the separation of photogenerated electrons and holes occurs through double charge transfer: an electron migrates from the CB of semiconductor 1 to the CB of semiconductor 2; a hole moves from the VB of semiconductor 2 to the VB of semiconductor 1. Electrons are accumulated in the CB of semiconductor 2, while holes accumulate in the VB of semiconductor 1, which facilitates efficient charge separation and reduced recombination [91,92]. Although traditional type II heterojunctions improve charge separation, they reduce the initial redox ability of electrons and holes.

The formation of S-scheme heterojunctions facilitates efficient charge separation without affecting the intrinsic redox ability of semiconductors [93]. To achieve successful construction of the S-scheme, the position of the CB and the Fermi level of the photocatalyst on which the reduction reaction proceeds must be higher than respective characteristics of the photocatalyst which runs the oxidation process. The internal electric field caused by the band deviation due to the difference in the Fermi levels of the two semiconductors enhances the charge mobility. The synthesis of two different semiconductors for subsequent integration into an S-scheme is considered rather difficult [94]. To date, the S-scheme heterojunction approach is considered to be the best choice for the development of photocatalysts with high charge separation efficiency and redox ability. Compared to S-scheme, the Z-scheme heterojunction photocatalysts have higher charge separation efficiency and are easier to synthesize [97]. Electrons from the CB of semiconductor 1 migrate through electron mediators (metals) to the VB of semiconductor 2 to recombine with photoinduced holes.

In the case of type I heterojunction photocatalysts [98], the redox process occurs on one semiconductor. The type II [99] heterojunctions significantly promote the separation of photoinduced electrons and holes, but have a weak redox ability that is detrimental for the photocatalytic reaction. Currently, the S-scheme approach is the most promising with regard to efficient charge separation and redox reactions [100]. However, the creation of S-scheme heterojunction photocatalysts is a difficult task that requires the structural modification of both semiconductors.

## 3. Synthesis of Graphitic Carbon Nitride to Provide Superior Photocatalytic Activity

In this section, the authors present their study performed at the Boreskov Institute of Catalysis and Nikolaev Institute of Inorganic Chemistry, that allowed the transition from the traditional precursor calcination method to the advanced synthesis of graphitic carbon nitride with improved textural and electronic characteristics, and creating highly active low-platinum catalysts for hydrogen production, which outperform the reaction rate of other catalysts reported in the literature [76,102,103,104].

### 3.1. Synthesis of Graphitic Carbon Nitride by Conventional Thermal Polycondensation

At the first step of the study, g-C_3_N_4_ was prepared by the traditional thermal condensation of organic precursors—melamine and dicyandiamide. The calcination was performed at varying temperatures (450, 500, 550 or 600 °C) and times (2 or 4 h) in air. In order to prevent charge recombination and increase the catalyst activity, 1–3 wt.% platinum was loaded by impregnating g-C_3_N_4_ with H_2_PtCl_6_, and then followed by reducing with a 2.5-fold excess of NaBH_4_. After several decantations, the obtained samples were dried at 50 °C for 4 h. The synthesized photocatalysts were characterized by XRD (Figure 3a) [76].

The main diffraction peak (2θ ~ 27°) can be assigned to the (002) plane; it determines the distance between the 2D layers of graphitic carbon nitride. The second peak at 2θ = 13° is due to the distance between tri-s-triazine units (Figure 3a). The specific surface area of photocatalysts increases significantly with an increase in the calcination temperature (Figure 3b). The samples calcined at temperatures of 450 and 500 °C demonstrate low specific surface areas of <10 m^2^·g^−1^; at calcination temperatures of 550 and 600 °C, the surface increases to 13–28 m^2^·g^−1^. Note that the specific surface area of g-C_3_N_4_ synthesized from melamine exceeds that of g-C_3_N_4_ obtained from dicyandiamide.

Figure 4 illustrates photocatalytic activity of the synthesized samples depending on the calcination conditions.

Among a large set of synthesized samples differing by precursors, calcination temperatures (450 to 600 °C) and times (2 or 4 h), the most active photocatalysts were selected (Figure 4). The higher the calcination temperature of the precursors, the more active the photocatalyst, most likely, due to the efficient (more complete) decomposition of melamine and dicyandiamide. Note that the correlation between the catalyst activity and the specific surface area is non-linear. Indeed, the catalyst, synthesized through melamine calcination at 600 °C for 2 h and having a lower specific surface area of 19.5 m^2^·g^−1^ appeared more active than a similar catalyst calcined for 4 h and having a larger surface area of 27.9 m^2^·g^−1^. This observation may be attributed to the optimal interplanar spacing and crystal size in the former sample.

Since platinum facilitates efficient separation of photoinduced charges, it was used as a cocatalyst. The 1 wt. % Pt additive provided a higher catalytic activity than the 3 wt. % one. As a result of the optimization of catalyst preparation procedure, the highest activity for hydrogen production—0.45 mmol g_cat_^−1^·h^−1^ (apparent quantum efficiency 0.6%)—showed the 1% Pt/g-C_3_N_4_ catalyst synthesized by melamine calcining at 600 °C for 2 h [76].

### 3.2. Synthesis of Graphitic Carbon Nitride from Melamine Hydrothermally Pretreated in Glucose

Based on the above results, melamine was chosen used as the precursor for catalyst synthesis using pretreatment in glucose. Melamine was suspended in an aqueous glucose solution and then heated in an autoclave at 180 °C for 12 h. The resulting material was heated in a furnace to 550 °C at a rate of 1°C/min, kept at this temperature for 1 h and then cooled to room temperature [102]. The synthesized material was used to deposit platinum (0.5 wt %) by various methods [102], one of which—sorption of Pt from nitratocomplex— has already proven itself in the case of deposition on TiO_2_ surface [30]. Besides platinum reduction with an excess of sodium borohydride, as described earlier, the following methods were used.

Photoreduction. Deaerated suspension consisting of carbon nitride, H_2_PtCl_6_, excess ethanol (20 vol.%) and water was illuminated by LED radiation with a wavelength of 380 nm (30 W).

Reduction of platinum nitratocomplexes. Carbon nitride was suspended in acetone and combined with an appropriate aliquot of the (Me_4_N)_2_[Pt_2_(μ-OH)_2_(NO_3_)_8_] acetone solution. The resulting suspension was stirred at room temperature for 12 h, then dried in an air stream. The obtained material was reduced in a hydrogen atmosphere (100–500 °C, heating rate 10 °C/min) for 1 h. The results of the samples characterization by XRD method proved that they had the structure of graphitic carbon nitride. The samples had the specific surface area (S_BET_) of ~25 m^2^·g^−1^, which is quite similar to respective value for the catalysts synthesized through standard procedure with precursor calcination, but the pore system was higher due to the use of glucose.

Obviously, the catalysts reduced by NaBH_4_ contain platinum in states 4+, 2+ and 0, while in photoreduced catalysts, platinum states are 2+ and 0. XPS analysis confirmed that for the samples synthesized with the use of a platinum nitratocomplex and reduced in hydrogen flow at temperatures ranging 100–500 °C, the intensity of peak at 70.7 eV increases with increasing temperature that corresponds to the gradual formation of metallic platinum from adsorbed Pt^2+^ species. At 400 °C, platinum is completely reduced to Pt^0^. Note also that at 500 °C, graphitic carbon nitride is completely destroyed, as proved by further experiments on the samples’ photocatalytic activity to produce hydrogen [102].

It has been shown that the sample calcined at 400 °C in H_2_ flow showed the highest catalytic activity (Figure 5a) [102]. The activity of the 0.5% Pt/g-C_3_N_4_ photocatalysts prepared using platinumnitratocomplexes increases as the temperature of the catalyst reduction in H_2_ elevates to 400 °C and sharply decreases as the temperature reaches 500 °C, because the g-C_3_N_4_ structure is destroyed at this temperature. The kinetic dependences for photocatalytic hydrogen evolution from an aqueous solution of TEOA were also obtained to compare various methods of platinum reduction (Figure 5b).

It should be noted that the catalyst calcined at 400 °C in hydrogen flow showed the highest activity among all other samples prepared with the proposed methods for platinum deposition and reduction [102].

Furthermore, the photocatalyst demonstrated high stability (Figure 6). After 7.5 h, the rate of photocatalytic hydrogen evolution with this catalyst decreased by only 8%. Note also that hydrogen production rate on this photocatalyst was 5.3 mmol g_cat_^−1^·h^−1^ (apparent quantum efficiency AQE = 3.0% (425 nm)). Thus, as compared to traditional melamine calcination method [76], a 12-fold increase in the catalyst activity and a double reduction in Pt content was achieved.

### 3.3. Synthesis of Graphitic Carbon Nitride from Supramolecular Melamine-Cyanuric Acid Adduct

The next method for the synthesis of graphitic carbon nitride consists in the thermolysis of supramolecular adduct formed by melamine and cyanuric acid [103]. Melamine and cyanuric acid were suspended in distilled water (300 mL); the suspension was heated at 90 °C for 12 h under constant stirring. The resulting material was heated in a furnace to 550 °C at a rate of 1°C/min and kept at this temperature for 1 h. The obtained g-C_3_N_4_ samples were loaded with platinum as described above (reduction of platinum nitratocomplex in hydrogen flow at temperatures 100–400 °C [102]).

Graphitic carbon nitride synthesized by this method demonstrated an interesting property – the structural erosion and formation of new pores upon calcination of the samples with adsorbed platinum complexes at a temperature of 400 °C [103]. Thermogravimetric studies in hydrogen atmosphere showed that the decomposition temperature of graphitic carbon nitride synthesized by this method decreases when (Me_4_N)_2_[Pt_2_(OH)_2_(NO_3_)_6_] complexes are adsorbed on its surface; for 0.5 wt.% Pt/g-C_3_N_4_ this decrease is more significant than for 0.1 wt.% Pt/g-C_3_N_4_. The decomposition of g-C_3_N_4_ releases ammonia and methane. Note also that the specific surface area of photocatalysts increases with increasing Pt content as well. Indeed, 0.1 wt.% Pt/g-C_3_N_4_ (T = 400 °C) has the specific surface area of 90 m^2^·g^−1^; 0.5 wt.% Pt/g-C_3_N_4_ (Т = 400 °С)—289 m^2^·g^−1^, while unmodified carrier—60 m^2^·g^−1^. Figure 7 illustrates the formation of a developed porous structure upon calcination of the (Me_4_N)_2_[Pt_2_(OH)_2_(NO_3_)_6_]/g-C_3_N_4_(IV) samples in hydrogen flow.

It was shown that the proposed g-C_3_N_4_ modifying method, which includes the synthesis of graphitic carbon nitride from the supramolecular melamine-cyanuric acid adduct, the deposition of platinum from the nitratocomplex by sorption method, followed by reduction in H_2_ flow at 400 °C, provided photocatalytic hydrogen production under the action of visible light at a rate of 11 mmol g_cat_^−1^·h^−1^ (apparent quantum efficiency of 6.7%) [103], which exceeds respective values reported in the literature.

Additionally, an interesting dependence was observed for these samples. Catalytic activity per 1 g of Pt noticeably increased with increasing platinum content (0.01, 0.05, and 0.1 wt.% Pt). However, as the mass fraction of Pt increased further from 0.1% to 0.5%, the respective activity value decreased from 8.8 to 2.3 mmol g_Pt_^−1^·h^−1^. At the same time, the rate of hydrogen production per 1 g of catalyst for the sample 0.5% Pt/g-C_3_N_4_ was about 20% higher than that value for catalyst 0.1% Pt/g-C_3_N_4_ (Figure 8a–c). Thus, 0.1 wt.% Pt loading is the optimum for efficient production of hydrogen. The apparent quantum efficiencies of hydrogen production were calculated for different wavelengths (action spectrum) (Figure 8d). The maximum quantum efficiency was 6.6% at a wavelength of 425 nm. The action spectrum of the photocatalyst agrees well with the Kubelka–Munk curve F(R) calculated from the diffuse reflectance spectrum of the 0.5% Pt/g-C_3_N_4_ (IV) photocatalyst.

According to the data of the catalyst characterization by physicochemical methods, the treatment with hydrogen of (Me_4_N)_2_[Pt_2_(OH)_2_(NO)_6_]/g-C_3_N_4_ composites results in platinum transition into a metallic state followed by catalytical hydrogenation of the g-C_3_N_4_ to form methane and ammonia, resulting in an increase in the g-C_3_N_4_ specific surface area and branching its porous structure. The observed high values of hydrogen production rate are associated with the proposed approach for the synthesis of photocatalysts based on graphitic carbon nitride. Compared to traditional thermal calcination of the precursor, thermolysis of the supramolecular adduct formed by melamine and cyanuric acid facilitates uniform removal of ammonia groups. Subsequent platinum depositing on the material followed by calcination in a hydrogen flow at 400 °C provides both the complete platinum reduction to a metallic state, and a significant increase in the specific surface area (up to 289 m^2^·g^−1^ in the case of 0.5%Pt/g-C_3_N_4_), because platinum burns off the surface of graphic carbon nitride, which induces the formation of new pores and porous structure branching. It should be emphasized also, that the proposed method promotes uniform and tight depositing of platinum nanosized (~1 nm) particles on the g-C_3_N_4_ surface) [103].

### 3.4. Composites Cd-Zn Sulfide Solid Solution/g-C_3_N_4_

To provide efficient charge separation and, as a consequence, an increase in the catalyst activity, two schemes were proposed to synthesize composites based on g-C_3_N_4_, Pt, and solid solutions of Cd and Zn sulfides. In Scheme 1, Cd and Zn sulfides were deposited on the surface of g-C_3_N_4_ obtained by the traditional calcination of melamine at 600 °C for 2 h followed by platinum depositing through chemical reduction method. In Scheme 2, platinum was deposited first, and then cadmium and zinc sulfides were loaded [104].

It was shown that in the sample 1%Pt/20%Cd_0.8_Zn_0.2_S/g-C_3_N_4_ synthesized according to Scheme 1, small particles of Pt are deposited on the surface of Cd-Zn sulfide solid solutions. In the sample 20%Cd_0.8_Zn_0.2_S/1%Pt/g-C_3_N_4_ synthesized by Scheme 2, large platinum clusters are located at the g-C_3_N_4_/Cd_1−x_Zn_x_S interface [104].

All samples synthesized by Scheme 2 were more active than those synthesized by Scheme 1 (Figure 9). It is assumed that within the framework of these Schemes, various mechanisms of charge transfer are realized. In samples 1%Pt/y%Cd_1−x_Zn_x_S/g-C_3_N_4_ (Scheme 1), the migration of holes and electrons proceeds by type II heterojunction mechanism. Upon illumination, electrons migrate from the CB of g-C_3_N_4_ to the CB of Cd_1−x_Zn_x_S, and then to Pt, where the photocatalytic reduction reaction occurs. In samples y%Cd_1−x_Zn_x_S/1%Pt/g-C_3_N_4_ (Scheme 2), the S-scheme heterojunction is probably realized. At the same time, electrons from the g-C_3_N_4_ CB, possessing high reducing ability, and holes from the Cd_0.8_Zn_0.2_S VB, possessing high oxidizing ability, participate in redox reactions on the photocatalyst surface. Catalyst 20%Cd_0.8_Zn_0.2_S/1%Pt/g-C_3_N_4_ demonstrated the highest value of catalytic activity amounting to 2.52 mmol g_cat_^−1^·h^−1^. Besides, Scheme 2 photocatalysts are more stable than Scheme I photocatalysts [104].

For comparison, the samples containing g-C_3_N_4_ synthesized as described in the above Section 3.2 and Section 3.3, and 0.5 wt.% Pt deposited from nitratocomplex, were loaded with 20 wt.% Cd-Zn sulfide solid solution according to Scheme 2. These catalysts appeared more active as compared to those based on g-C_3_N_4_ obtained by the thermal polycondensation of the melamine precursor, and demonstrated hydrogen productivity of 6.3 and 10.2 mmol g_cat_^−1^·h^−1^, respectively.

### 3.5. Summary Data on the Photocatalysts Based on g-C_3_N_4_

Table 1 summaries the data on the activity of g-C_3_N_4_ based photocatalysts for hydrogen production obtained by the authors of this review. It is seen that the photocatalyst modification with Pt noticeably improved the catalyst activity. The catalyst based on g-C_3_N_4_ synthesized from glucose-pretreated melamine showed a 12-times higher photocatalytic activity in hydrogen production than the sample prepared by the traditional method. The catalyst based on g-C_3_N_4_ synthesized by thermal polycondensation of a supramolecular complex melamine-cyanuric acid, followed by platinum reduction from the nitratocomplex in hydrogen flow at 400 °C, provided a 24-fold increase in the hydrogen production rate as compared to that of the catalyst synthesized through thermal polycondensation of melamine and depositing platinum by the reduction of H_2_PtCl_6_ with NaBH_4_. It should be emphasized that, besides considerable increase in the catalytic activity, the twice-reduced platinum loading in the catalysts was achieved (0.5 instead of 1 wt.%).

Composite systems based on g-C_3_N_4_, Cd_1−x_Zn_x_S, and Pt were also considered. A significant increase in activity (5.6-fold) showed only the catalyst on the base of g-C_3_N_4_ obtained by thermal polycondensation of melamine. In the case of the other two methods (see Table 1), the increase in activity was either insignificant or not observed at all. This very interesting observation deserves more research and will be explained later.

Table 2 presents the data on catalytic activities in hydrogen production of various photocatalysts based on g-C_3_N_4_ reported in the literature and compares them with the results from Table 1. Clearly, the photocatalyst based on g-C_3_N_4_, synthesized from the melamine-cyanuric acid supramolecular adduct, with Pt cocatalyst, deposited from Pt-nitratocomplex and reduced in H_2_ at 400 °C, outperforms other catalysts in the photocatalytic production of H_2_. A comprehensive approach to the synthesis, that provided a high specific surface area of g-C_3_N_4_ and uniform distribution of Pt in the metallic state, allowed the superior catalyst performance in photocatalytic hydrogen production [103]. Also, as another interesting approach, the precipitation of Pt from the acidic solution of platinum hydroxide ([Pt(OH)_4_(H_2_O)_2_]) has been proposed. This approach is characterized by a simpler method of Pt deposition and is more green, since it does not require the isolation of complex salts and the use of organic solvents in the deposition process [105]. However, the rates of photocatalytic hydrogen production on photocatalysts synthesized by this method were slightly lower than in the case of using the nitratocomplex as a platinum precursor [103].

## 4. Conclusions

Over the past decade, graphitic carbon nitride has been recognized as a promising photocatalytic material and studied in sufficient detail. A lot of approaches to modify graphic carbon nitride were suggested; the most commonly used methods are described in this review. Analysis of up-to-day research data on this material proved that high rates of hydrogen evolution can be achieved by modifying the textural and electronic characteristics of graphic carbon nitride. The most promising approaches to solve this task include increasing the specific surface area, creating a branched pore system, and depositing the platinum group metals by various methods.

The authors of the review performed extensive research and proposed various approaches to the synthesis of graphitic carbon nitride, including traditional thermal polycondensation by melamine/dicyandiamide calcination, thermal polycondensation of melamine pretreated hydrothermally with glucose, and thermal polycondensation of the supramolecular melamine-cyanuric acid adduct. Additionally, various methods for depositing the platinum co-catalyst have been suggested. Platinum was deposited from H_2_PtCl_6_ and platinum nitratocomplexes and subsequently reduced by various methods, namely, by adding NaBH_4_, photoreduction, or in H_2_ flow. It was shown that the best method is to obtain graphitic carbon nitride from the supramolecular melamine-cyanuric acid complex in combination with the platinum depositing from Pt nitratocomplex by sorption method followed by reduction in H_2_ flow. This approach allowed the reaching a 11 mmol g_cat_^−1^·h^−1^ catalytic activity in hydrogen evolution, that significantly exceeds that value of photocatalysts obtained with the use of other combinations of g-C_3_N_4_ syntheses and platinum deposition methods. Currently, the proposed materials possess one of the highest performances in photocatalytic production of H_2_ as compared to recently published data.

In general, high values of photocatalytic activity of g-C_3_N_4_ based systems proposed by the authors of this review, and in other studies considered in the review, show obvious promises for up-scaling the process of photocatalytic hydrogen production towards practically and commercially valid level. However, to move to this level, it is necessary to solve not only the synthesis of active materials, but, for example, the simplification of the synthesis process, the use of environmentally friendly solvents, and the reduction of the content of noble metals in photocatalysts for hydrogen production, which was clearly shown in this study.

## Figures and Tables

**Figure 1 ijms-24-00346-f001:**
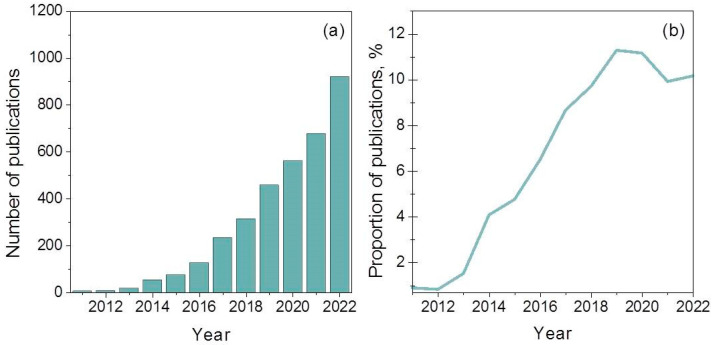
(**a**) Number of annual publications using “g-C_3_N_4_ photocatalytic H_2_ production” as a keyword since 2011. (**b**) Proportion between the number of publications with keywords “g-C_3_N_4_ photocatalytic H_2_ production” and “photocatalytic H_2_ production” based on data of Science Direct (Elsevier).

**Figure 2 ijms-24-00346-f002:**
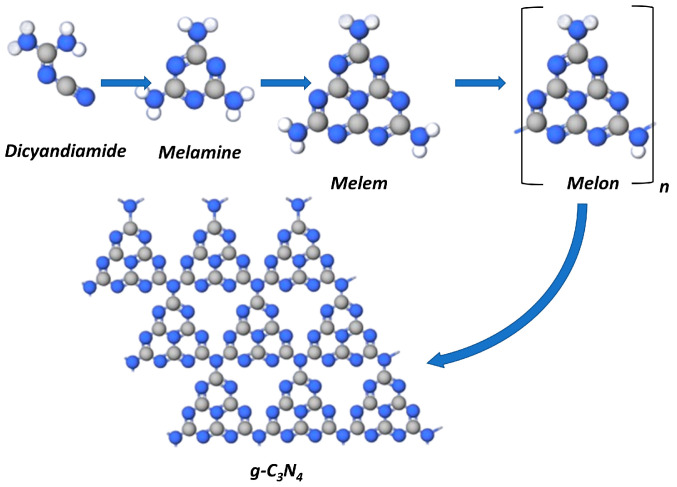
Thermal condensation of dicyandiamide to form g-C_3_N_4_.

**Figure 3 ijms-24-00346-f003:**
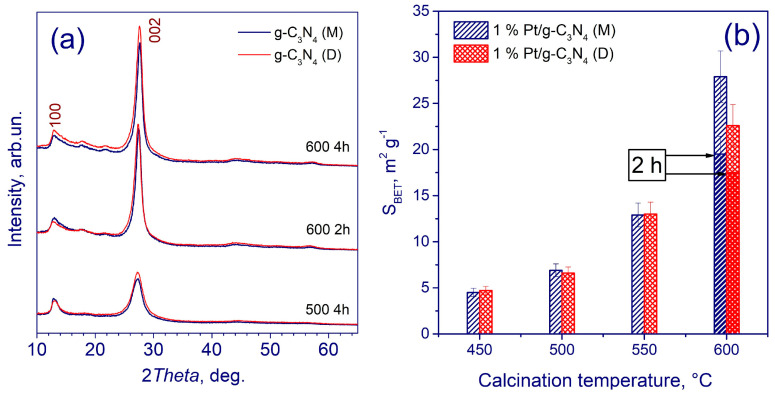
(**a**) XRD patterns of g-C_3_N_4_. (**b**) Specific surface area photocatalysts obtained from melamine (M) and dicyandiamide (D). Adapted with permission from Ref. [76]. 2021, Elsevier.

**Figure 4 ijms-24-00346-f004:**
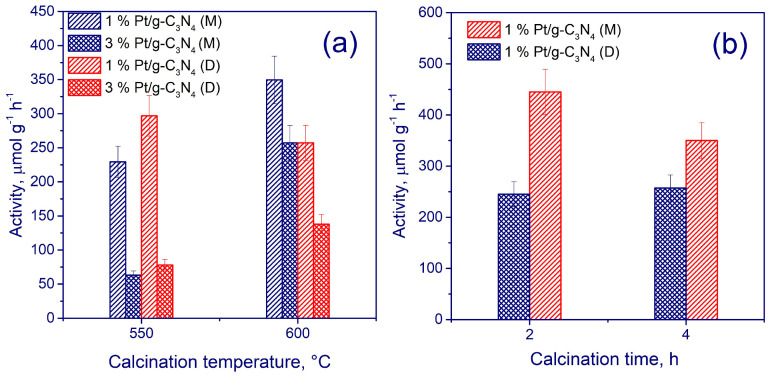
(**a**) Catalytic activity of samples synthesized at 550 and 600 °C for 4 h. (**b**) Catalytic activity of samples synthesized for 2 and 4 h at 600 °C using photocatalysts obtained from melamine (M) and A dicyandiamide (D). Adapted with permission from Ref. [76]. 2021, Elsevier.

**Figure 5 ijms-24-00346-f005:**
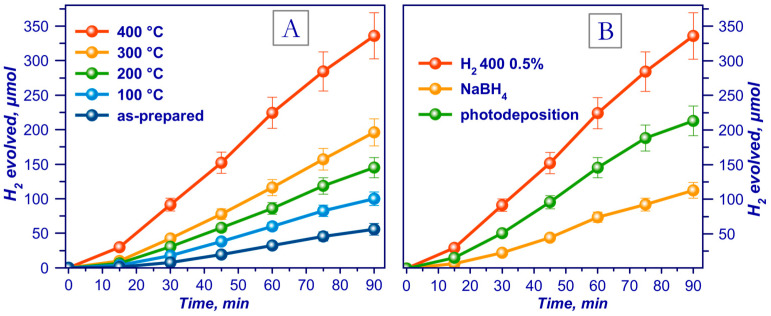
Kinetic curves of the photocatalytic evolution of hydrogen from an aqueous solution of triethanolamine (TEOA) over (**A**) catalysts reduced in H_2_ flow at different temperatures, and (**B**) using photocatalysts reduced by various methods. Adapted with permission from Ref. [102]. 2022, Elsevier.

**Figure 6 ijms-24-00346-f006:**
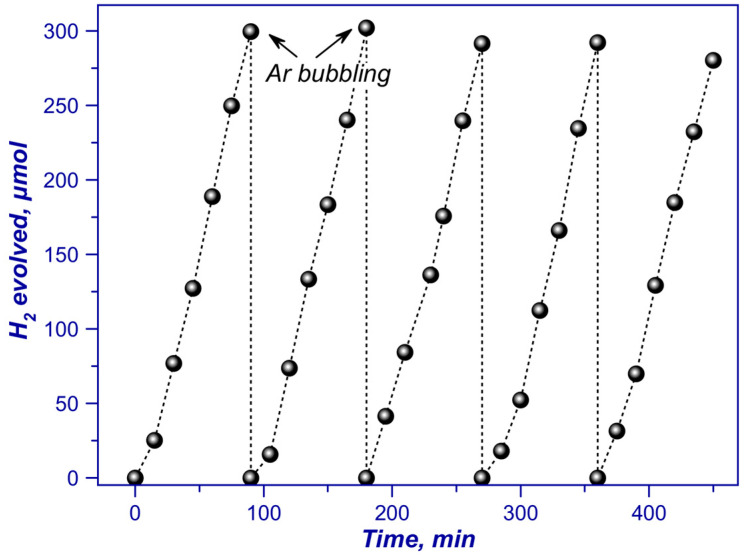
Stability of the 0.5% Pt/g-C_3_N_4_ (T = 400 °C) photocatalyst in five 1.5 h cycles of hydrogen production. Adapted with permission from Ref. [102]. 2022, Elsevier.

**Figure 7 ijms-24-00346-f007:**
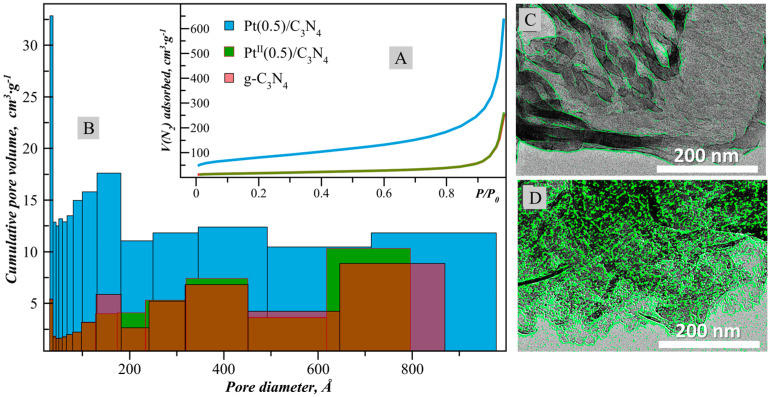
(**A**) Adsorption isotherms; (**B**) pore size distribution for g-C_3_N_4_, 0.5% Pt/g-C_3_N_4_ (IV) without treatment and 0.5% Pt/g-C_3_N_4_ (IV) after reduction in H_2_ at 400 °C. (**C**) TEM images of photocatalysts 0.5% Pt/g-C_3_N_4_ (no treatment) and (**D**) 0.5% Pt/g-C_3_N_4_ (Т = 400 °С). Reprinted with permission from Ref. [103]. 2022, Elsevier.

**Figure 8 ijms-24-00346-f008:**
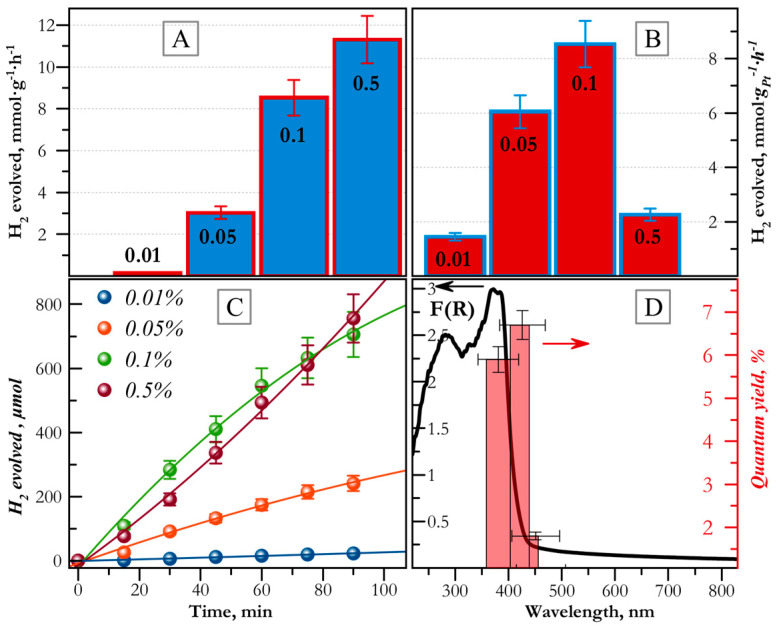
Activity of the Pt_x_/g-C_3_N_4_ photocatalysts in the reaction of hydrogen evolution (λ = 425 nm) calculated by (**A**) per gram of catalyst; (**B**) per gram of Pt. (**C**) Kinetic dependences of the H_2_ production reaction under the action of light for Pt_x_/g-C_3_N_4_ catalysts. (**D**) Dependence of the quantum efficiency on the wavelength for the Pt_0.1_/g-C_3_N_4_ catalyst. Adapted with permission from Ref. [103]. 2022, Elsevier.

**Figure 9 ijms-24-00346-f009:**
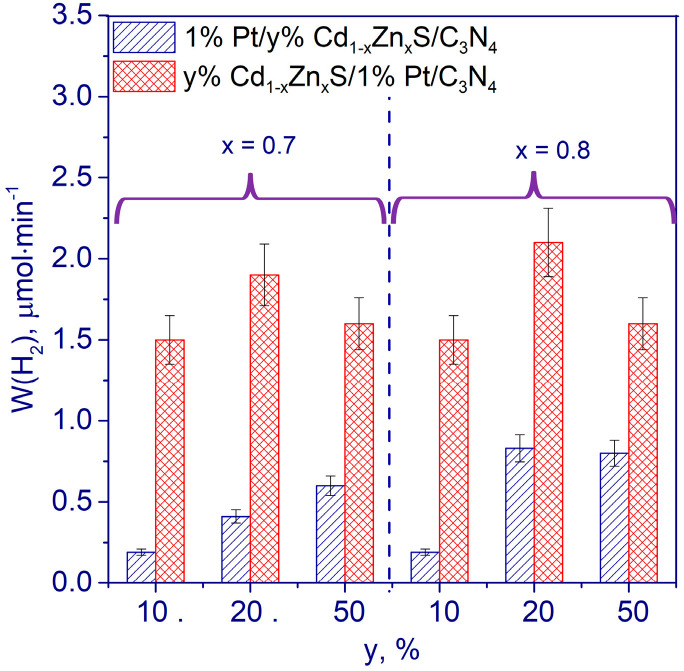
Photocatalytic activity of samples 1%Pt/y%Cd_1−x_Zn_x_S/g-C_3_N_4_ and y%Cd_1−x_Zn_x_S/1% Pt/g-C_3_N_4_. Adapted from [104].

**Table 1 ijms-24-00346-t001:** Comparison of catalytic activities of various photocatalysts described in Chapter 3.

g-C_3_N_4_ Synthesis	Pt Deposition			Composites y%Cd_0.8_Zn_0.2_S/x%Pt/g-C_3_N_4_
	H_2_PtCl_6_		(Me_4_N)_2_[Pt_2_(OH)_2_ (NO_3_)_6_]	
	Reduction by NaBH_4_	Photoreduction	H_2_ Reduction at 400 °C	
Thermal polycondensation of melamine	0.45	0.87	1.80	2.52
Thermal polycondensation of melamine pretreated in glucose by the hydrothermal method	1.80	4.20	5.30	6.30
Thermal polycondensation of the supramolecular complex melamine-cyanuric acid	3.00	4.90	11.0	10.2

* All data in the table has a dimension mmol g_cat_^−1^·h^−1^.

**Table 2 ijms-24-00346-t002:** Comparison of characteristics of the Pt/g-C_3_N_4_ based photocatalysts for hydrogen production reported in the literature and the most active photocatalysts described in Table 1.

№	Catalyst	Pt, wt.%	Catalyst Mass, mg	Sacrificial Agent	Synthetic Procedure of Pt	Light Source	H_2_ Evolution Rate, µmol min^−1^	Catalytic Activity, µmol g_cat_^−1^·h^−1^	AQY, %	Ref.
1	0.15-NVCN x—mass KOH heating in N_2_ atmosphere	0.5	30	20 vol.% methanol	H_2_PtCl_6_ photodeposition in situ	300W Xe lamp λ > 420 nm	1.66	3310	8.60	[74]
2	g-C_3_N_4_	0.5	10	20 vol.% TEOA	H_2_PtCl_6_ Ar plasma	Xe-lamp	0.19	1151	-	[75]
3	g-C_3_N_4_ nanosheets	1.25	100	5 vol. % TEOA pH = 9	H_2_PtCl_6_ ethylene glycol reduction	300W Xe lamp λ > 420 nm	0.75	451	-	[106]
4	CNC0.1 0.1%—weight ratio between urea and glucose	1.0	50	15 vol. % TEOA	H_2_PtCl_6_ photodeposition in situ	350 W Xe lamp λ ≥ 420 nm	0.18	213	0.90	[107]
5	S-doped g-C_3_N_4_ S-CN	0.3	100	15 vol. % methanol	H_2_PtCl_6_ photodeposition in situ	300W Xe lamp	1.24	742	-	[108]
6	VCN heating in NH_3_ atmosphere	3.0	100	15 vol. % TEOA	H_2_PtCl_6_ photodeposition in situ	300W Xe lamp λ > 420 nm	5.51	3300	-	[109]
7	HCN-3h heating in H_2_ atmosphere	3.0	50	15 vol. % TEOA	H_2_PtCl_6_ photodeposition in situ	300W Xe lamp	3.58	4300	-	[110]
8	ultrathin O-doped g-C_3_N_4_ nanosheets	3.0	5.0	10 vol. % TEOA	H_2_PtCl_6_ photodeposition in situ	300W Xe lamp λ > 400 nm	3.16	3790	-	[111]
9	mesoporous g-C_3_N_4_	0.5	100	10 vol. % isopropanol	H_2_PtCl_6_ photodeposition in situ	medium pressure Hg arc lamp (125W)	3.35	2010	2.72	[112]
10	NiS/g-C_3_N_4_-30 30—time irradiation	-	5	10 vol. % TEOA	-	300W Xe lamp λ > 400 nm	0.27	3300	1.25	[113]
11	30 wt.% CdS/g-C_3_N_4_	-	20	10 vol.% methanol	-	300W Xe lamp λ > 420 nm	1.22	3670	2.03	[114]
12	25 wt.% ZnCo_2_S_4_/C_3_N_4_	-	20	20 vol.% TEOA	-	300W Xe lamp λ > 420 nm	2.21	6620		[115]
13	g-C_3_N_4_	0.5	50	10 vol. % TEOA	Reduction of chemisorbed Pt nitrato complex with H_2_ at 400 °C	425-nm LED	9.40	11300	6.70	[103]
14	g-C_3_N_4_	0.1	50	10 vol. % TEOA	Reduction of chemisorbed Pt nitrato complex with H_2_ at 400 °C	425-nm LED	7.10	8500	5.01	[103]
15	g-C_3_N_4_	0.5	50	10 vol. % TEOA	Reduction of chemisorbed ([Pt(OH)_4_(H_2_O)_2_] with H_2_ at 400 °C	425-nm LED	7.10	8500	5.01	[105]

## Data Availability

The data presented in this study are available on request from the corresponding author.

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
