# Peer review of "Comprehensive Review on g-C_3_N_4_-Based Photocatalysts for the Photocatalytic Hydrogen Production under Visible Light"

_ijms, 2022, doi:10.3390/ijms24010346_

Round 1

Reviewer 1 Report

This publication is a comprehensive overview of the synthesis of graphite-like carbon nitride. The review is well structured, modern publications are considered. However, a number of remarks arise: 1. I believe that at least 100 articles should be considered in a review for IJMS. 2. Table 2 should be supplemented with other examples, for example with heterostructures.

Author Response

Thank you for the comments and helping us to improve our manuscript. We have acted on the suggested changes in the revised manuscript. The corrections proposed by Referee 1 are marked in yellow.

Answer to the remarks of Referee 1:

  1. I believe that at least 100 articles should be considered in a review for IJMS.

Author reply: Thank you for this recommendation; we have expanded the number of references of our review.

  1. Table 2 should be supplemented with other examples, for example with heterostructures.

Author reply: The references have also been added to the Table 2 to compare data in the Table 2 with composite materials based on g-C3N4.

Reviewer 2 Report

The authors have written a review article entitled “Comprehensive review on g-C3N4-based photocatalysts for the photocatalytic hydrogen production under visible light”. The manuscript is quite interesting, well framed, and based on the graphic carbon nitride as a photocatalyst for hydrogen production under visible light is presented. The authors have described the concept to a greater extent but the manuscript still needs some Minor corrections before publishing in the International Journal of Molecular Sciences

I advise the authors to consider the following points when revising their manuscript.

Comment 1: Abstract should clearly discuss the problem statement, so it should be revised.

Comment 2: The manuscript needs to be checked for typographical/ grammatical, superscript, and subscript errors.

Comment 3: The introduction section needs to be improved. The authors should cite some significant references in this section and also highlight the graphic carbon nitride as a photocatalyst in the introduction section.

Gao, R., et al. "Graphitic carbon nitride (g-C3N4)-based photocatalytic materials for hydrogen evolution." Frontiers in chemistry 10 (2022).

Cao, Shaowen, and Jiaguo Yu. "g-C3N4-based photocatalysts for hydrogen generation." The journal of physical chemistry letters 5.12 (2014): 2101-2107.

Alaghmandfard, Amirhossein, and Khashayar Ghandi. "A Comprehensive Review of Graphitic Carbon Nitride (g-C3N4)–Metal Oxide-Based Nanocomposites: Potential for Photocatalysis and Sensing." Nanomaterials 12.2 (2022): 294.

Xia, Pengfei, et al. "Synthesis of g-C3N4 from Various Precursors for Photocatalytic H2 Evolution under the Visible Light." Crystals 12.12 (2022): 1719.

Comment 4: Include the structured graphical abstract in the revised manuscript to attain a broad readership.

Comment 5: The authors need to add the graphical representation figure and explanation of the publication trend (2000-2020) in the field of graphic carbon nitride as a photocatalyst.

Comment 6: The applications of g-C3N4 with an improved photocatalytic performance by developed several strategies such as adjusting pH, morphology control, participation of co-catalyst, and construction of heterojunction. So, these mentioned strategies should be included and discussed in the revised version of the manuscript.

Comment 7: Figure 3 quality is very poor, so improve the resolution of figure 3.

Comment 8: Lines 219-224 provide the reference in the paragraph.

Comment 9: Please check whether the authors have already obtained the copyright permissions for the presented Figures.

Comment 10: The conclusion section needs to be revised with a future directions, also based on the critical discussion made in the manuscript. 

Comment 11:  The homogeneity of the reference section needs to be maintained. So please check and revise accordingly to the journal's instructions. 

Author Response

Thank you for the comments and helping us to improve our manuscript. We have acted on the suggested changes in the revised manuscript. The corrections proposed by Referee 2 are marked in blue.

Answer to the remarks of Referee 2:

  1. Abstract should clearly discuss the problem statement, so it should be revised

Author reply: We have revised the abstract and discussed the problem statement.

  1. The manuscript needs to be checked for typographical/grammatical, superscript, and subscript errors.

Author reply: The manuscript was checked for typographical/grammatical, superscript and subscript errors, errors were corrected.

  1. The introduction section needs to be improved. The authors should cite some significant references in this section and also highlight the graphic carbon nitride as a photocatalyst in the introduction section.

Author reply: Proposed references were added in the manuscript.

  1. Include the structured graphical abstract in the revised manuscript to attain a broad readership.

Author reply: Graphical abstract was included in the manuscript.

  1. The authors need to add the graphical representation figure and explanation of the publication trend (2000-2020) in the field of graphic carbon nitride as a photocatalyst.

Author reply: This figure has been added to the text as Figure 1.

  1. The applications of g-C3N4 with an improved photocatalytic performance by developed several strategies such as adjusting pH, morphology control, participation of co-catalyst, and construction of heterojunction. So, these mentioned strategies should be included and discussed in the revised version of the manuscript.

Author reply: Paragraph 2.1.3 was added to the manuscript describing the effect of pretreatment with acids or alkalis on the surface of g-C3N4.

  1. Figure 3 quality is very poor, so improve the resolution of figure 3.

Author reply: We have changed the resolution of Figure 3. In new version, its designed as Figure 4.

  1. Lines 219-224 provide the reference in the paragraph

Author reply: The references have been added to the manuscript.

  1. Please check whether the authors have already obtained the copyright permissions for the presented Figures.

Author reply: The copyright have been obtained for the presented Figures.

  1. The conclusion section needs to be revised with a future directions, also based on the critical discussion made in the manuscript. 

Author reply: We revised conclusion chapter and discussed the future directions.

  1. The homogeneity of the reference section needs to be maintained. So please check and revise accordingly to the journal's instructions.

Author reply: The references have been corrected in accordance with the journal's instructions.

Reviewer 3 Report

In this review, "Comprehensive review on g-C3N4-based photocatalysts for the photocatalytic hydrogen production under visible light," Graphic carbon nitride as a visible-light photocatalyst for hydrogen production is discussed.

The essential methods are synthesizing g-C3N4 and increasing catalyst hydrogen production by altering composition, structure, and photocatalytic properties.

This review discusses g-C3N4-based photocatalyst production and ways to optimize its structural and photocatalytic capabilities.

Next, the authors describe the synthesis of graphitic carbon nitride and its photocatalysts, which produce hydrogen efficiently and reliably.

The causes of g-C3N4 materials were discussed. The topics are well-organized and thoroughly discussed. Therefore, I would recommend it be published in the International Journal of Molecular Sciences after the following minor issues are addressed:

  1. Please revise the resolution of Figure 3.
  2. There are some extra spaces. For some examples, check page 4, line 143; page 5, line 183; page 7, line 251.
  3. Please check page 4, line 118 (formation)
  4. For the review article, the abbreviation should be treated more carefully. The authors should check again to ensure that they define the abbreviation when mentioned first in the article. Then use the abbreviation in the following part. In addition, the authors could refrain from using abbreviations if they mention them a few times in the article.

Author Response

Thank you for the comments and helping us to improve our manuscript. We have acted on the suggested changes in the revised manuscript. The corrections proposed by Referee 3 are marked in green.

Answer to the remarks of Referee 3:

  1. Please revise the resolution of Figure 3.

Author reply: Thank you for this recommendation; we have changed the resolution of Figure 3. In new version, its designed as Figure 4.

  1. There are some extra spaces. For some examples, check page 4, line 143; page 5, line 183; page 7, line 251.

Author reply: All extra spaces have been removed, for convenience these places are also highlighted in green in the text.

  1. Please check page 4, line 118 (formation)

Author reply: Word wrap has been removed.

  1. For the review article, the abbreviation should be treated more carefully. The authors should check again to ensure that they define the abbreviation when mentioned first in the article. Then use the abbreviation in the following part. In addition, the authors could refrain from using abbreviations if they mention them a few times in the article.

Author reply: For use, we left one abbreviation triethanolamine – TEOA. In figure 3 and figure 4, we used the designations M and D as melamine and dicyandiamide, added a decoding in the captions to the figures.
